# Stress in Metastatic Breast Cancer: To the Bone and Beyond

**DOI:** 10.3390/cancers14081881

**Published:** 2022-04-08

**Authors:** Catarina Lourenço, Francisco Conceição, Carmen Jerónimo, Meriem Lamghari, Daniela M. Sousa

**Affiliations:** 1Instituto de Investigação e Inovação em Saúde (I3S), Universidade do Porto, 4200-135 Porto, Portugal; clourenco@i3s.up.pt (C.L.); francisco.conceicao@i3s.up.pt (F.C.); lamghari@ineb.up.pt (M.L.); 2INEB—Instituto Nacional de Engenharia Biomédica, Universidade do Porto, 4200-135 Porto, Portugal; 3Cancer Biology and Epigenetics Group, Research Center of IPO Porto (CI-IPOP)/RISE@CI-IPOP (Health Research Network), Portuguese Oncology Institute of Porto (IPO Porto)/Porto Comprehensive Cancer Center (Porto.CCC), 4200-072 Porto, Portugal; carmenjeronimo@ipoporto.min-saude.pt; 4ICBAS-UP—School of Medicine & Biomedical Sciences, University of Porto, 4050-313 Porto, Portugal; 5Department of Pathology and Molecular Immunology—ICBAS-UP, 4050-313 Porto, Portugal

**Keywords:** breast cancer, metastasis, stress, sympathetic nervous system, adrenergic receptors

## Abstract

**Simple Summary:**

Breast cancer is the most common cancer affecting women of all ages worldwide. In spite of the encouraging advances made in early diagnosis, 10% of breast cancer patients are still affected with metastatic breast cancer at the time of diagnosis. The available therapeutic options are predominantly palliative, and thus this unfavourable prognosis is associated with a low survival rate. Intriguingly, stress has been shown to promote the growth of breast tumours and the incidence of metastasis. Herein, we describe the contribution of the sympathetic hyperactivation induced by stress to the progression of breast cancer and its dissemination to distant organs, specifically to the bone, but also to the lung, liver and brain. The putative sympathetic adrenergic signalling mechanisms responsible for this modulation are also summarised. The knowledge gathered highlights the therapeutic potential of targeting sympathetic signalling to tackle cancer progression and metastasis.

**Abstract:**

Breast cancer (BRCA) remains as one the most prevalent cancers diagnosed in industrialised countries. Although the overall survival rate is high, the dissemination of BRCA cells to distant organs correlates with a significantly poor prognosis. This is due to the fact that there are no efficient therapeutic strategies designed to overcome the progression of the metastasis. Over the past decade, critical associations between stress and the prevalence of BRCA metastases were uncovered. Chronic stress and the concomitant sympathetic hyperactivation have been shown to accelerate the progression of the disease and the metastases incidence, specifically to the bone. In this review, we provide a summary of the sympathetic profile on BRCA. Additionally, the current knowledge regarding the sympathetic hyperactivity, and the underlying adrenergic signalling pathways, involved on the development of BRCA metastasis to distant organs (i.e., bone, lung, liver and brain) will be revealed. Since bone is a preferential target site for BRCA metastases, greater emphasis will be given to the contribution of α2- and β-adrenergic signalling in BRCA bone tropism and the occurrence of osteolytic lesions.

## 1. Introduction

Breast cancer (BRCA) is the most frequently diagnosed cancer in women worldwide and is expected to represent around 25% of all new cancer cases diagnosed in females. In 2020, more than 2 million people were diagnosed with BRCA, with Europe accounting for nearly 24% of all new cancer cases [1,2]. Despite the large number of new cases, the mortality rate has been slowly decreasing in developed countries with the implementation of earlier diagnosis and the improvement of adjuvant therapies [3,4,5,6]. Nevertheless, metastatic BRCA still affects 6–10% of women at the time of diagnosis and presents a 5-year survival rate of 27% [6,7,8].

The development of metastasis and its prediction can be dictated by specific risk factors, such as grade, nodal involvement and size of the tumour. However, these factors do not predict the specific sites or patterns of metastasis, characteristic of BRCA tumours. Interestingly, it has been hypothesised that the primary tumour can provide insight about the organ that BRCA-disseminating cells eventually home to, partaking in the possibility to influence the therapeutic and survey strategies for each patient since the time of primary diagnosis. Although BRCA subtypes present a known organotropism [9], this process remains still largely unexplained.

In addition, patients diagnosed with BRCA, undergoing surgery or therapy, are at a higher risk of feeling emotional stress [10,11,12,13,14]. These symptoms may lead up to a psychiatric disorder, such as anxiety or depression, and can develop several years after the diagnosis of the disease. The link between cancer and emotional disorders has been suggested since the ancient times, however the nature of this association has only started to be revealed during the last two decades [14,15,16,17,18,19]. Although a few studies demonstrate an association between stress-induced sympathetic hyperactivation and the incidence of cancer and dissemination [14], there seems to be a stronger and more consistent relationship between psychological factors and the progression of already-existing tumours [20,21,22,23,24,25]. Nevertheless, greater emphasis has been granted to the use of drugs targeting stress-induced signalling pathways in cancer initiation and progression [26], and more importantly in metastatic BRCA [27].

Fortunately, the nature of this association between stress and cancer progression is being gradually uncovered in order to better understand how chronic stress and the sympathetic tone influences the metastatic cascade and how it can affect the destination of BRCA cells. In this review, BRCA heterogeneity will be discussed, focusing on its effects on metastatic site predisposition. Likewise, the consequences of a sympathetic hyperactivation, owing to chronically stressed conditions, on BRCA metastasis will also be reviewed.

## 2. The Disease Fundamentals of BRCA

### 2.1. BRCA Molecular Subtypes

The heterogeneity and complexity of BRCA has long been noted by accessing histologic samples and patients’ outcomes [28,29]. The development of molecular profiling techniques has further ensured this heterogeneity, and it is now possible to classify BRCA within, at least, three main subtypes: luminal, HER2-enriched (HER2^+^) and triple-negative breast cancer (TNBC). As detailed in Figure 1, each molecular subtype has different biological characteristics, including risk factors, prognosis, response to therapies and a preferential metastatic site [30,31,32,33,34]. Luminal tumours express receptors for oestrogen (ER) and progesterone (PR) hormones, and are divided into two subtypes: luminal A and luminal B [35]. The other subtypes represent hormonal-negative tumours and usually portray a worse prognosis than the luminal subtypes. While the HER2-enriched subtype illustrates the tumours that have a high expression of the HER2 gene and other genes related to its pathway [36], the TNBC subtype, often referred to as basal-like tumours, mimics the expression profile of the myoepithelial cells and usually lacks the expression of both hormonal receptors (HR) and HER2. Moreover, not all TNBC are basal-like tumours [37,38] and a new subclass of TNBC lacking cell–cell adhesion and tight junction’s markers (e.g., claudins) has emerged, denominated the claudin-low subtype [39,40,41]. Overall, the novel information provided by molecular profiling techniques has allowed to better understand each breast tumour subtype, generating new treatment approaches to be used as individualised therapies, which will be translated in the improvement of BRCA patient outcomes.

### 2.2. Patterns of Metastatic BRCA

In the end, metastasis is a disorganised multifactorial process where the ability for a primary tumour to metastasize to a specific organ depends on a variety of factors, including the cancer cell type, the primary organ and the microenvironment of the secondary site [42]. The intrinsic characteristics of cancer cells and the cellular and cytokine profile of the tissue of origin dictate how these cells will migrate, survive and proliferate. The tissue microenvironment to which metastatic cells eventually home also plays a significant role in this process. Most importantly, the interaction between the primary organ and the secondary site commands the success of metastasis [43,44].

Organ-specific metastasis was firstly described by Paget in 1889 in the “seed and soil” hypothesis, where he, after evaluating BRCA patient autopsies, stated that “in cancer of the breast, the bones suffer in a special way” [45]. The author proposed that these patterns were due to the seed dependency (cancer cell) for the soil (environment factors in the new organ), suggesting, therefore, that the distribution of metastatic sites is not a random act.

In general, BRCA cells commonly metastasize to bone, lungs, liver and brain [46]. Interestingly, the majority of studies showed that the luminal subtypes, tumours with positive ER and PR expression, metastasize preferentially to the bone, taking a longer time to relapse. On the other hand, tumours with negative HR expression, such as TNBC subtypes, present a smaller tropism to bone and are usually present in the brain and lungs, and rapidly recur. The HER2^+^ subtype mostly metastasizes to visceral organs (Figure 1) [9,47,48]. The median overall survival of metastatic BRCA patients ranges from approximately 1 year for metastatic TNBC, to 5 years for HR and HER2^+^ BRCA subtypes [49]. Thus, each BRCA subtype not only displays specific primary tumour characteristics, such as aggressiveness and response to treatments, but can also exhibit different metastatic behaviour. This knowledge is very important since it can help in the development of follow-up and surveillance strategies for newly diagnosed patients, allowing for different options of adjuvant therapies. However, the characteristics and the mechanisms that determine the location of BRCA metastatic spreading are still largely unknown, and thus more research is required in this field.

## 3. The Sympathetic Nervous System (SNS) Response to Stress in BRCA

### 3.1. Stress and the Activation of the SNS

Behavioural stress has been pointed out as an accelerator of cancer progression. Indeed, stress is a complex process where both psychosocial and environmental factors trigger a cascade of information-processing pathways in the central and peripheral nervous system [50]. Both the hypothalamic-pituitary-adrenal (HPA) axis and the SNS systems have been implicated in cancer, supported by an increasing body of studies linking the “fight-or-flight” stress response of SNS mediators with cancer progression [21,22,51,52,53]. For this reason, the investigation on the role played by the SNS on cancer biology has been largely encouraged.

By controlling involuntary body functions, SNS virtually regulates all human organs. The activation of SNS leads to the release of catecholamines, such as norepinephrine (NE) and epinephrine (E), which regulate these functions through two possible pathways. On one hand, there is the localised release of NE from the sympathetic nervous terminals that directly innervate the target organs, along with the co-release of sympathetic non-adrenergic neurotransmitters (e.g., neuropeptide-Y and ATP), whereas the other pathway mainly involves the systemic release of NE and E (in a proportion of 20:80, respectively) by the adrenal glands to the circulation [54]. It is noteworthy that growing evidence suggests that the local release of NE from SNS nerve terminals is the dominant driving force in the sympathetic control of cancer progression [55,56,57].

The sympathetic neuro-mediators, NE and E, bind to different adrenergic receptors (ADRs) that are G protein-coupled receptors (GPCR). The ADRs show distinct patterns of tissue distribution, have diverse functions and can even originate opposite actions, ultimately regulating physiological homeostasis. The two main classes of ADR are the alpha (α-ADR) and the beta (β-ADR), which can be further subdivided into nine subtypes: α1-ADR (α1A, α1B and α1D), α2-ADR (α2A, α2B and α2C) and β-ADR (β1, β2, β3) [51,58]. The α1- and β-ADRs are excitatory receptors, while the α2-ADRs present inhibitory pathways [54].

During the SNS acute “fight-or-flight” responses, E and NE levels can increase by 10 times. This causes rapid physiological changes in respiratory, cardiovascular, muscular, immune and neural systems, increasing the blood flow to muscles and lungs, preparing the body for alert situations. The catecholamines levels return to baseline in a very short amount of time (20–60 min), and therefore the activation of the acute stress responses is considered adaptive [53]. On the other hand, in chronic stress conditions, the physiological systems are exposed to glucocorticoids and catecholamines for long periods of time. This increased exposure leads to a deterioration of health conditions, such as increased risk of infections and cardiac diseases, decreased wound healing and eventually death [59]. Importantly, in the case of cancer, catecholamines have the potential to induce a panoply of physiologic effects, both deleterious and beneficial, termed the cancer catecholamine conundrum. These variable effects were described to be dependent on several factors, such as: (i) catecholamine concentration in the blood, (ii) exposure time, (iii) physical activity, (iv) activation of nine different ADR and (v) the duration and recurrence of the stress, among others [60]. Overall, a correlation between stress and tumorigenesis/cancer progression endures in different types of tumours, namely BRCA [61].

### 3.2. Effects of Stress in Primary BRCA

Since it has been reported that the stress response is partly mediated by the activation of ADRs, several studies have confirmed that BRCA cancer cells and BRCA tissues generally express α- or β-ARs [62,63,64]. The ADRs expression and its associated correlations with BRCA are summarised in Table 1.

**Table 1 cancers-14-01881-t001:** A summary of the expression of ADRs in BRCA and associated correlations.

ADR Subtype		Major Correlations	Refs
Alpha (α)	α1B	⇑ Expression in high-grade, HER2^+^, luminal-like cancers Inverse association with luminal markers Poor cancer-specific survival ⇑ Tumour recurrence	[65]
	α2A	Inversely associated with HER2 status Association with ER status Blockade ⇑ distant metastasis-free survival	[66,67]
	α2C	⇑ Expression in high-grade tumours Inversely associated with HER2 status Inverse association with PR status Blockade ⇑ distant metastasis-free survival ⇑ Tumour size and metastatic relapse	[65,66,67]
Beta (β)	β1	⇑ Expression in luminal-like and HER2^+^ cancers	[68]
	β2	Low-grade, luminal-like (ER^+^) cancers ⇑ Disease-free survival in ER^+^/PR^+^ positive Blockade ⇑ distant metastasis-free survival	[65,66,67]
	β3	⇑ Expression in luminal-like and HER2^+^ cancers	[68]

Abbreviations: ER—oestrogen receptor; HER2—human epidermal growth factor 2; PR—progesterone receptor.

The most described α-ADR in BRCA is the α2-ADR subtype, which was shown to cause increased proliferation of BRCA cell lines [69,70], tumour growth [70,71,72], metastasis [73,74,75] and chemoresistance of cancer cells [76]. Interestingly, not only can E and NE act directly on tumour-express α2-ADR, but SNS activation can also influence tumour growth and metastasis through α2-ADR present on stromal cells of the tumour microenvironment, even when specific BRCA cell lines did not express functional α-ADRs [74].

β-ADRs were the first ADRs to be described in BRCA tissues [77] and cell lines [78], with the β2-ADR subtype being the most expressed of the β-ADRs. According to its stimulatory G-protein nature, β-ADR promotes tumour growth in various studies using BRCA cell lines [68,79,80,81,82]. Interestingly, however, other studies have described an opposite effect on tumour cell proliferation, and thus the role of β-ADR activation on BRCA proliferation remains controversial [70,83]. Besides proliferation, β-ADR has mainly been associated with increased BRCA cells’ migration [84,85,86,87,88] and metastasis [82,85,89,90,91]. Studies also provided evidence for a direct role of the β-adrenergic signalling pathway in the acceleration of tumour angiogenesis [92,93]. However, this role of β-AR in promoting cancer progression is not observed in every study [94], and therefore is still not confirmed.

The prognostic significance of α- and β-ADRs expression in BRCA tumours has also been explored, suggesting a possible role for targeted therapy using ADR antagonists. While α-ADR has been described to be expressed in poor-prognosis tumours, β2-ADR-expressing tumours are mostly associated with good prognosis [65,66,67]. Furthermore, a retrospective study observed that HER2^+^ tumours expressing β2-ADR presented a significantly lower disease-free survival rate and lymph node metastasis incidence [62]. Interestingly, besides the putative effect of ADR expression on BRCA prognosis, the ADR expression profile seems to be dependent on the BRCA subtype. For instance, luminal-like tumours strongly express β2-ADR, while α-ADRs, such as α1B- and α2C-ADR, are overexpressed in basal-like breast tumours [65].

It is important to note that ADR expression in BRCA is still not consistent. Besides different expression profiles between tissues and cell lines, the same cell line in different studies may not express the same ADRs, and this is probably due to the use of distinct quantification methodologies. For instance, MCF-7, a luminal A BRCA cell line, has been shown to express β2-ADR in some studies at the mRNA level, while in another study, β2-ADR was suggested to be negatively expressed [69,81]. Additionally, MCF-7 has been shown to exhibit the highest β2-ADR protein expression level among the tested BRCA cell lines [86]. Thus, further investigation is required to ensure correct characterisation of the adrenergic profile of commonly used cell lines and match it with clinical human biopsies. Even though the presence and the effects of ADRs in BRCA remains to be clarified, these studies clearly suggest that SNS activation, through the signalling of ADRs, influences BRCA progression and metastasis. Furthermore, the link between stress and BRCA progression was strengthened when pharmaco-epidemiological studies showed that the use of β-blockers, at the time of diagnosis, was associated with improved survival, decreased tumour invasion, metastasis, recurrence and mortality [95,96,97,98]. These studies present some limitations, such as the limited size of the analysed patient cohort, and the benefits of β-blocker usage on improved survival was not replicated in other epidemiologic studies [99,100]. Additionally, the role of β-blockers (e.g., propranolol), as a neoadjuvant therapy, has been assessed in a small number of clinical trials, yet these studies did not deliver major conclusive results (NCT01847001; NCT02596867). Interestingly, a phase II randomised controlled trial highlighted the benefits of using propranolol in BRCA patients, since only one week of treatment was associated with decreased expression of pro-metastatic biomarkers [101]. Nevertheless, further pre-clinical and clinical studies clarifying the importance of ADRs on different subtypes of BRCA could potentiate the development of novel therapeutic strategies.

### 3.3. Effects of Stress in Metastatic BRCA

#### 3.3.1. Targeting the Bone Microenvironment

BRCA bone metastatic foci are often characterised by the formation of osteolytic lesions, where the interaction between tumour cells and the bone niche leads to the establishment of a vicious cycle of bone destruction and, subsequently, complications such as fractures, hypocalcaemia and severe bone pain [102,103,104]. When BRCA-disseminated cells arrive at the bone microenvironment, they disrupt the intricate cascade of events that regulate bone remodelling to ultimately favour bone resorption. In fact, at the site, BRCA cells can directly activate osteoclasts (the bone-resorbing cells) or act indirectly through osteoblasts (the bone-forming cells) by stimulating osteoblast-derived receptor activator of NF-κB ligand (RANKL), which is a master regulator of osteoclastogenesis [105]. Additionally, BRCA cells can also inhibit the differentiation and adhesion of osteoblasts, increase their apoptosis and delay collagen synthesis, thus impairing osteoblasts’ capacity to fully replace the resorbed bone. The over-activation of osteoclast bone resorption will lead to the release of ionised calcium and growth factors entrapped in the bone matrix, which will further stimulate the growth and survival of cancer cells. This self-perpetuating cycle leads to bone loss and tumour growth and is designated as the “osteolytic vicious cycle” of BRCA bone metastasis [106].

Bone appears to be a preferential organ for homing of BRCA metastasis [107,108,109], and therefore, most studies regarding the effect of SNS activation in BRCA have explored the bone niche. Besides cancer cells, various bone stromal cell types, such as osteoclasts [110,111,112], osteoblasts [113,114] and mesenchymal stem cells (MSCs) [115,116], also express ADRs, supporting the premise that the peripheral sympathetic neurons play an important role in bone remodelling and other bone physiologic processes [117,118,119,120].

Some studies have explored the effects of SNS activation on bone remodelling diseases, where increased incidence of fractures and decreased bone mass were observed [121,122,123,124,125,126]. Osteoclasts express ADRs, and SNS activation has been shown to promote bone loss, by directly affecting bone resorption in mice [127]. Furthermore, β-blockers [128] and α2-ADRs agonists [112] inhibited the mRNA expression of the osteoclast-related genes such as TRAP and cathepsin K, and decreased the number of TRAP-positive multi-nucleated osteoclasts. In humans, NE was described to inhibit osteoclast-precursor cell proliferation on osteoclast-precursor cells and increased osteoclast maturation and TRAP activity [129]. These studies suggest that ADRs are involved in the regulation of osteoclastogenesis by directly affecting osteoclast activity. However, the majority of the studies mainly focus on the role of osteoblasts, suggesting that local NE release and binding to osteoblastic β2-ADR leads to inhibition of bone formation and stimulation of bone resorption, mainly due to augmented RANKL expression [130,131,132].

Several preclinical and epidemiologic studies have shown that β-blockers, drugs that inhibit β-adrenergic signalling and are commonly used to treat hypertension, have been linked with reduced BRCA metastasis and improvement of the patient survival [95,96,97,98,133,134,135,136]. Furthermore, a vast number of in vitro and in vivo studies have explored the presence and role of α- and β-ADRs in BRCA, highlighting an association between chronic SNS stimulation and BRCA progression [137]. Adrenergic signalling seems to either directly affect BRCA cells [67,73,74,75,90,138] or indirectly influence cells in the pre-metastatic niche to facilitate disease progression [89,93,139,140,141,142,143,144]. Studies performed in other common sites for metastasis also suggest a major indirect effect of SNS in stromal cells, facilitating the migration and colonisation of cancer cells in these organs during chronically stressed states [89,140,142,144].

During the last decade, multiple studies that focused on BRCA showed that SNS activation can modulate BRCA bone metastasis. Besides the direct stimulation of BRCA ADRs, some studies have shown that SNS can also act on bone marrow stromal cells to indirectly promote the colonisation of bone by metastatic BRCA cells. SNS activation, via β-AR signalling in osteoblasts, induced RANKL production, which promoted BRCA metastasis to bone via its pro-migratory effect on RANK-expressing BRCA cells. Moreover, blocking SNS activation with a β-blocker inhibited the stimulatory effect of sympathetic activation on bone metastasis [139]. The activation of β2-AR expressed in osteoblasts also seems to play a crucial role in mediating the SNS signals in bone, through the production of vascular endothelial growth factor (VEGF) [143]. Moreover, β2-ADR stimulation in osteoblasts triggers the release of soluble factors, such as IL-1β, that favour BRCA cell engraftment within the skeleton, by the upregulation of E- and P-selectin expression by endothelial cells [141]. Overall, the role that these cytokines play in bone metastasis suggests that when chronic stress activates the SNS, the bone marrow microenvironment is transformed into a more favourable tissue for the establishment of metastasis [145].

#### 3.3.2. Other Metastatic Sites: Beyond the Bone

Even though the skeleton is the preferential site for metastasis in a vast majority of BRCA tumours, other organs such as lungs, brain, lymphatic nodes and liver can also be targeted and colonised by metastatic BRCA cells. Although there is a scarce number of studies exploring the role of SNS in these organs, the influence of chronic stress is beginning to be noticed, with special interest in the β2-adrenergic signalling pathway.

For instance, β-adrenergic signalling activation was shown to promote lung metastatic colonisation by BRCA cells. Treatment with the β-adrenergic antagonist propranolol supressed the stress-induced metastasis, while stimulation of β-adrenergic signalling with isoproterenol (ISO) increased the number of monocytes and infiltration of macrophages into the pre-metastatic lung. Thus, under chronic stress, β-ADR stimulation promotes metastatic homing and seeding of circulating BRCA cells through remodelling of the pre-metastatic lung microenvironment [140]. Previously, Shakhar et al. demonstrated that the systemic β-adrenergic stimulation in rats suppressed the activity of NK cells and caused an increase in tumour cell retention in the lungs, that was later translated in increased lung metastases. Importantly, the use of a β-adrenergic antagonist reversed these effects [144]. Additionally, clinical trials have shown that propranolol combined with etodolac, a COX-2 inhibitor, decreased the risk of lung metastasis. This combination also enhanced tumour clearance from the lungs and improved long-term recurrence-free survival rates, showing its potential in limiting post-operative immunosuppression and metastatic progression [146]. However, other studies have demonstrated that treatment with a β2-ADR agonist decreased the number and size of BRCA lung metastases [94]. Despite the controversial results, these studies highlight the role of β-ADRs in lung metastasis development. Finally, although there are only a few studies addressing the role of α-ADR signalling on lung metastasis, it was previously demonstrated that treatment with dexmedetomidine, an α2-ADR agonist, leads to increased tumour-cell retention and growth of BRCA lung metastases in vivo [74,75].

Besides the action of SNS on stromal and immune cellular components of the pre-metastatic niche, another study demonstrated that an augmented tumour invasiveness and metastasis in lung and lymph nodes was caused by neuroendocrine signalling directly affecting BRCA cells [89]. Additionally, using the RNA interference methodology, Chang et al. showed that β2-ADR knockdown in BRCA cells reduced tumour cell invasion in vitro and significantly reduced the impact of stress on lung and lymphatic metastasis in vivo [90].

Interestingly, chronic stress was found to be a pathophysiological regulator of lymphatic remodelling in cancer, facilitating BRCA dissemination to nearby lymph nodes. Through tumour-derived VEGFC and macrophage-derived COX-2, chronic stress restructured lymphatic networks within and around tumours, providing pathways for tumour cell escape. Inhibition of SNS signalling using propranolol blocked chronic stress effects and reduced metastasis to lymph nodes. Conversely, SNS activation with ISO was sufficient to increase lymph node metastasis. Furthermore, in agreement with the pre-clinical evidence discussed above, β-blocker use reduced the risk of lymph node and distant metastasis in a group of BRCA patients [142].

Among the different subtypes of BRCA, the prevalence of brain metastasis in TNBC is the highest [147]. The SNS can also modulate brain metastasis progression since a retrospective study showed that perioperative β-blockade was associated with decreased cancer recurrence in stage II BRCA patients [138]. Moreover, TNBC metastatic cells presented increased β2-ADR mRNA and protein expression levels relative to TNBC from the primary tumour. When a β2-ADR agonist was used to mimic stress conditions in vitro, TNBC brain-metastatic cells exhibited increased cell proliferation and migration, which was abrogated by propranolol. In addition, propranolol also decreased the establishment of brain metastases in vivo [138].

Liver is also another common metastatic site in BRCA, and even though the effects of chronic stress in BRCA liver metastasis have not been assessed, SNS activation was previously shown to influence tumour dissemination to the liver in other cancer types, i.e., colon cancer [148]. In a study with socially isolated mice, an enhancement of liver metastasis from a colon carcinoma cell line was observed. Chronic stress was used as a model of endogenous SNS activation characterised by elevated levels of catecholamines, NE and E. These stressed mice developed metastatic foci at earlier time points, presented a decreased survival time and displayed worse chemotherapeutic responses than control mice. Interestingly, β-blocker treatment reversed these effects [149].

Chronic stress was also associated with increased infiltration of tumour-associated macrophages into the primary tumour and increased the expression of metastatic genes [148]. In addition, propranolol inhibited proliferation and induced apoptosis in liver cancer cells, confirming that the SNS activation may have a role in liver cancer, particularly through the β-adrenergic signalling pathway [150]. Since similar mechanisms might be in play in BRCA, further studies exploring the effects of SNS activation in BRCA liver metastasis are warranted.

## 4. Concluding Remarks and Future Perspectives

Several reports support the hypothesis that chronic SNS activation plays a critical role in the establishment of metastatic BRCA, specifically in the bone microenvironment. The information gathered so far on this association is described in Figure 2. Nevertheless, the results obtained so far are scarce, in some cases contradictory, and the mechanisms remain poorly understood. Thus, more studies need to be performed, particularly, a better characterisation of the adrenergic profile of the different BRCA subtypes, and its influence in metastasis of chronically stressed patients, would be of great importance. Additionally, the role of other adrenergic signalling pathways (besides β-adrenergic) should also be clarified.

It is now clear that unveiling the mechanisms behind the correlations between stress, BRCA and metastasis may allow for the development of specific therapies for metastatic BRCA. The use of α- and β-blockers in BRCA animal models suggests a role for pharmacologic inhibition of ADRs signalling in helping to control the metastatic progression [73,118,139]. Moreover, since cancer diagnosis, surgery and associated treatment is a highly stressful experience, potentially worsening the progression of the disease, the general use of ADR-blockers systemically (specifically β-blockers) as an adjuvant therapy has the ability to improve the effectiveness of cancer treatment [26,151]. In addition, the use of alternative non-pharmacological approaches has also been suggested to reduce the sympathetic activity via the modulation of local neuronal activity and improve BRCA patients’ survival outcomes [151]. Overall, further investigation will be needed to determine the best time for stress-management interventions [14] and to further analyse the clinical benefits of such approaches.

## Figures and Tables

**Figure 1 cancers-14-01881-f001:**
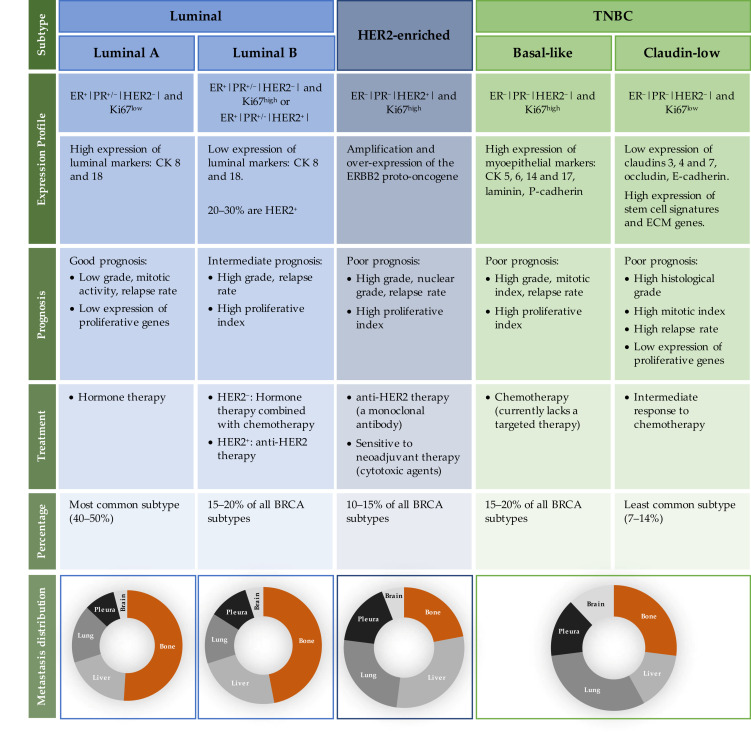
Intrinsic characteristics of each BRCA subtype, in terms of receptor expression profile, prognosis, treatment, percentage of incidence and metastasis distribution. Abbreviations: BRCA—breast cancer; CK—cytokeratin; ECM—epithelial to mesenchymal transition; ER—oestrogen receptor; ERBB2—human epidermal growth factor receptor-type 2; HER2—human epidermal growth factor 2; Ki 67—antigen Ki-67; PR—progesterone receptor; TNBC—triple-negative breast cancer.

**Figure 2 cancers-14-01881-f002:**
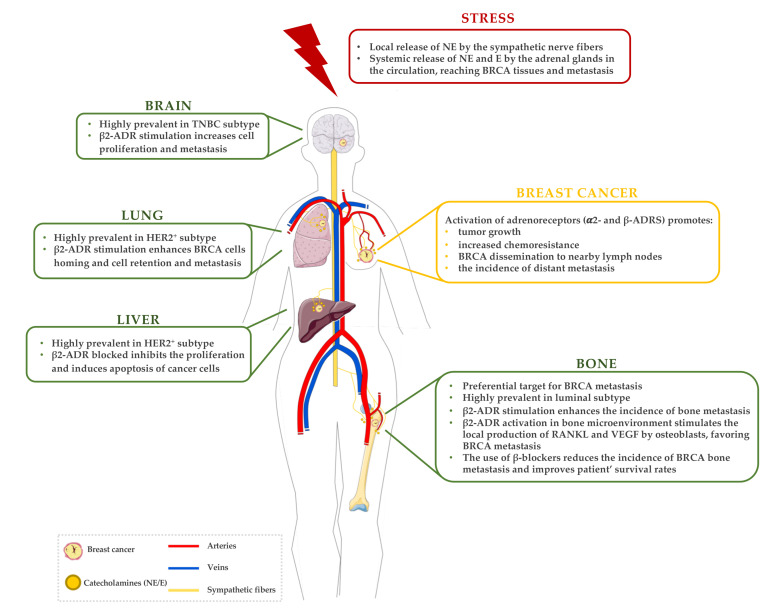
A brief summary of the established correlations between stress, BRCA and metastases. In a stress situation, catecholamines (NE and E) are released by the sympathetic fibres and sympathoadrenal system. These catecholamines stimulate the activation of adrenergic receptors, locally in BRCA primary tissue and also by affecting the BRCA metastatic organs. Abbreviations: ADR—adrenergic receptor; BRCA—breast cancer; E—epinephrine; HER2—human epidermal growth factor 2; NE—norepinephrine; RANKL—receptor activator of NF-κB ligand; TNBC—triple-negative breast cancer; VEGF—vascular endothelial growth factor.

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
