# Peer review of "Stress in Metastatic Breast Cancer: To the Bone and Beyond"

_cancers, 2022, doi:10.3390/cancers14081881_

Round 1
Reviewer 1 Report
In their manuscript Lourenço et al. provided insight into the role of stress in metastatic breast cancer.
They focused on the role of epinephrinee and norepinephrine and their corresponding receptors in the processes associated with the development of metastases in breast cancer patients. The manuscript is well written, but I have a few comments.
Comments:
In addition to norepinephrine, sympathetic nerve endings release also neuropeptide Y. Please desribe the effect of this neurotransmitter on metastases in breast cancer. Also describe the role of other sympathetic co-transmitters in this process (e.g. ATP).
Page 2, lines 56-64: Please mention the most recent articles describing the role of stress in cancer initiation and progression (Eckerling, A., Ricon-Becker, I., Sorski, L., Sandbank, E., Ben-Eliyahu, S., 2021. Stress and cancer: mechanisms, significance and future directions. Nat Rev Cancer.; Mravec, B., Tibensky, M., Horvathova, L., 2020. Stress and cancer. Part I: Mechanisms mediating the effect of stressors on cancer. J Neuroimmunol 346, 577311.; 10.3389/fcell.2021.777018; 10.3389/fonc.2021.738252; 10.3390/cancers13225816; 10.3390/ijms22116115; 10.1080/2162402X.2021.2004659)
Please mention the role of the sympathoadrenal system in the mechanisms related to tumor surgery-induced dissemination of cancer cells and development of metastasis (see papers of Ben-Eliyahu et al.).
Page 5, lines 173-183: It should be noted that there are several factors that determine the effects of catecholamines on cancer, including the concentration of catecholamines in the blood, the duration of stress,… (for more details see 10.1016/j.trecan.2021.10.005).
Page 5, lines 185-189: The expression of catecholamine biosynthetic enzymes in cancer tissue take a place mainly in immune cells - these catecholamines are not related to catecholamines released by the sympathoadrenal system.
Page 6, lines 235-237: Please mention current clinical trials using propranolol in the treatment of breast cancer (see https://www.clinicaltrials.gov/)
Page 9, lines 370-394: Please mention that reducing β-adrenergic signaling may also increase the effectiveness of cancer treatment (10.3390/ijms22116115). It is also worth mentioning when to start treatment and other approaches that reduce the activity of the sympathoadrenal system in cancer patients in order to maximize the beneficial effects of these interventions in breast cancer patients. (10.3390/ijms21217958)
Author Response
The authors would like to acknowledge the two reviewers for the feedback on this manuscript.
Both reviewers raised concerns related to relevant aspects of the manuscript, and we believe that the alterations made improved the quality of this manuscript.
The authors were pleased to address all the comments.
Please find the alterations highlighted in yellow in the revised version of the manuscript, and the response to each of the comments below.
REVIEWER #1
In their manuscript Lourenço et al. provided insight into the role of stress in metastatic breast cancer.
They focused on the role of epinephrinee and norepinephrine and their corresponding receptors in the processes associated with the development of metastases in breast cancer patients. The manuscript is well written, but I have a few comments.
Comments:
- In addition to norepinephrine, sympathetic nerve endings release also neuropeptide Y. Please describe the effect of this neurotransmitter on metastases in breast cancer. Also describe the role of other sympathetic co-transmitters in this process (e.g. ATP).
Reply 1. The authors thank the reviewer for the suggestion. Indeed, the role of NPY and other sympathetic (non-adrenergic) co-neurotransmitters in cancer progression are extremely relevant. However, the focus of this review has been to illustrate the role of adrenergic signaling in breast cancer metastasis. Still, NPY and ATP has been briefly mentioned in Page 4, lines 147-148.
“…directly innervate the target organs along with the co-release of sympathetic non-adrenergic neurotransmitters (e.g. neuropeptide-Y and ATP); whereas…”
- Page 2, lines 56-64: Please mention the most recent articles describing the role of stress in cancer initiation and progression (Eckerling, A., Ricon-Becker, I., Sorski, L., Sandbank, E., Ben-Eliyahu, S., 2021. Stress and cancer: mechanisms, significance and future directions. Nat Rev Cancer.; Mravec, B., Tibensky, M., Horvathova, L., 2020. Stress and cancer. Part I: Mechanisms mediating the effect of stressors on cancer. J Neuroimmunol 346, 577311.; 10.3389/fcell.2021.777018; 10.3389/fonc.2021.738252; 10.3390/cancers13225816; 10.3390/ijms22116115; 10.1080/2162402X.2021.2004659)
Reply 2. The authors agree with the Reviewer’s comment. These studies have been included has requested. Please refer to Page 2, lines 56-67.
- Please mention the role of the sympathoadrenal system in the mechanisms related to tumor surgery-induced dissemination of cancer cells and development of metastasis (see papers of Ben-Eliyahu et al.).
Reply 3. This has also been briefly mention in the manuscript. Please refer to Page 2, lines 56-67.
- Page 5, lines 173-183: It should be noted that there are several factors that determine the effects of catecholamines on cancer, including the concentration of catecholamines in the blood, the duration of stress,… (for more details see 10.1016/j.trecan.2021.10.005).
Reply 4. Although this recent article is not accessible for download for these authors, we have agreed to include the information requested and available on the journal webpage and previous work published:
“Importantly, in the case of cancer, catecholamines have the potential to induce a panoply of physiologic effects both deleterious and beneficial, termed as cancer catecholamine conundrum. These variable effects were described to be dependent on several factors, such as: i) catecholamine concentration in the blood; ii) exposure time; iii) physical activity; iv) activation of 9 different ADR; v) the duration and recurrence of the stress; among others [60].”
Please refer to Page 5, lines 171-176.
Page 5, lines 185-189: The expression of catecholamine biosynthetic enzymes in cancer tissue take a place mainly in immune cells - these catecholamines are not related to catecholamines released by the sympathoadrenal system.
Reply 5. The authors appreciate the reviewer’s comment. For the sake of clarity, and since the enzymes involved in the biosynthesis of catecholamines are not mention throughout the manuscript nor the focus of the review, the authors have deleted that part of the sentence: “as well as enzymes involved in catecholamine biosynthesis, such as tyrosine-hydroxylase (TH), phenylethanolamine N-methyltransferase (PNMT) or dopamine beta-hydroxylase (DBH)” (Page 5, lines 179-182 ).
Page 6, lines 235-237: Please mention current clinical trials using propranolol in the treatment of breast cancer (see https://www.clinicaltrials.gov/)
Reply 6. We thank the Reviewer for the nice suggestion. Although, several clinical trials have been mentioned along the manuscript, as recommended, the following sentence was added to Page 6, lines 230-235.
" Additionally, the role of β-blockers (e.g. propranolol), as a neoadjuvant therapy has been assessed in a small number of clinical trials, yet these studies did not deliver major conclusive results (NCT01847001; NCT02596867). Interestingly, a phase II randomized controlled trial highlighted the benefits of using propranolol in BRCA patients, since only one-week of treatment was associated with decreased expression of pro-metastatic biomarkers [101].".
- Page 9, lines 370-394: Please mention that reducing β-adrenergic signaling may also increase the effectiveness of cancer treatment (10.3390/ijms22116115). It is also worth mentioning when to start treatment and other approaches that reduce the activity of the sympathoadrenal system in cancer patients in order to maximize the beneficial effects of these interventions in breast cancer patients. (10.3390/ijms21217958)
Reply 7. The concluding remarks have been fully reformulated to meet with the reviewer’s #1 and #2 comments. Please refer to Page 9, lines 370-391.
Reviewer 2 Report
The authors intended to correlate the sympathetic activity causing stress to the development of bone metastasis in breast cancer patients. The topic is interesting and was not covered in the last years.
I have some concerns about the review composition.
First, the Breast cancer classification is unnecessary in Cancer Journal. There are thousands of publications describing the genetic and molecular classifications in BC. A summary could be sufficient for the purpose of the review.
The authors must be focused on the neurological changes in stress. In this review a separate chapter on stress and sympathetic activity is missing. Additionally, the correlation between stress, sympathetic activity and bone metastasis must be exposed with more precision.
Table 1 is useful
Figure 1 which contains a table is not clear and the figure legend do not coincide with the content. Abbreviations must be included in all figures/table’s legends.
In page 4 the firs sentence is unnecessary, it can be added to the background.
The concluding remarks are what the authors thought to expose in this review but in this review is not clearly described. Please intend to connect between the 3 branches more clearly.
An illustrated scheme could be useful for this matter.
Author Response
The authors would like to acknowledge the two reviewers for the feedback on this manuscript.
Both reviewers raised concerns related to relevant aspects of the manuscript, and we believe that the alterations made improved the quality of this manuscript.
The authors were pleased to address all the comments.
Please find the alterations highlighted in yellow in the revised version of the manuscript, and the response to each of the comments below.
REVIEWER #2
The authors intended to correlate the sympathetic activity causing stress to the development of bone metastasis in breast cancer patients. The topic is interesting and was not covered in the last years.
I have some concerns about the review composition.
- First, the Breast cancer classification is unnecessary in Cancer Journal. There are thousands of publications describing the genetic and molecular classifications in BC. A summary could be sufficient for the purpose of the review.
Reply: The authors appreciate the observation raised by the reviewer. Indeed, a full chapter describing the Breast cancer classification might be overstated in Cancers Journal, especially because the Figure 1 included in Chapter 2 already gives a broad overview on the disease. In line with the reviewer’s opinion, Chapter 2.1 has been summarized. Please refer to Page 2, lines 77-95.
- The authors must be focused on the neurological changes in stress. In this review a separate chapter on stress and sympathetic activity is missing. Additionally, the correlation between stress, sympathetic activity and bone metastasis must be exposed with more precision.
Reply 2: The authors agree with the opinion of the reviewer. The headlines and sub headlines of Chapter 3 of this manuscript have been reorganized in order to portray the reviewer’s point of view, namely:
“3. The sympathetic nervous system (SNS) response to stress in BRCA
3.1 Stress and the activation of SNS
3.2 Effects of stress in primary BRCA
3.3 Effects of stress in metastatic BRCA
3.3.1 Targeting the bone microenvironment
3.3.2 Other metastatic sites: beyond the bone”
- Table 1 is useful
Figure 1 which contains a table is not clear and the figure legend do not coincide with the content. Abbreviations must be included in all figures/table’s legends.
Reply 3: The authors agree with the reviewer. In line with the Reviewer’s concerns, the abbreviations have been either revised (Figure 1) or included (in the case of Table 1 and new Figure 2).
- In page 4 the first sentence is unnecessary, it can be added to the background.
Reply 4: The sentence has been excluded from the manuscript.
- The concluding remarks are what the authors thought to expose in this review but in this review is not clearly described. Please intend to connect between the 3 branches more clearly.
Reply 5: The concluding remarks have been fully reformulated to meet with the reviewer’s #1 and #2 comments. Please refer to Page 9, lines 370-391.
- An illustrated scheme could be useful for this matter.
Reply 6: A second figure has been include to better illustrate the state of knowledge on the correlations between Stress (SNS activation)/Breast Cancer/Metastasis. Please refer to Figure 2 on Page 10.
Round 2
Reviewer 2 Report
The authors corrected the manuscript based on the suggested comments.